# MissingBench-Verified: Probing Vision-Language Models' Inability to Detect Missing Object Parts

## Abstract

Vision Language Models (VLMs) are well known for hallucinating non-existent objects in images. Objects with missing parts present a unique challenge for VLMs, stemming from both real-world knowledge bias and the scarcity of such images in training data. We present MissingBench-Verified, a benchmark designed to evaluate a specific and practically relevant scenario: when vision-language models fail to recognize that an essential component of an object has been removed. Across ten leading models, we observe consistent and significant failure rates that persist even when external tool evidence explicitly contradicts the model's visual perception. Error analysis reveals that models frequently dismiss tool outputs, attribute missing regions to occlusion or framing artifacts, or confabulate object attributes to reconcile the contradiction. We further ask whether granting models access to image processing tools (e.g., cropping, contrast adjustment) enables autonomous inspection to resolve these failures. We find that existing mitigation strategies, including tool-assisted verification, autonomous visual reasoning, longer reasoning durations, and fine-tuning on an easier dataset, provide negligible improvement, indicating that this failure mode cannot be addressed through current prompting or post-hoc correction techniques. Our findings highlight a fundamental limitation of current VLM for inspection and monitoring tasks and underscore the need for architectural or training-level interventions that enable models to override internal expectations when confronted with contradictory evidence.

## 1 Introduction and Related Work

Vision Language Models (VLMs) are well known for hallucinating non-existent objects in images. This phenomenon was first observed in the pre-LLM era with image captioning models (Rohrbach et al., 2019) and has received increasing attention in work on modern VLMs (Zhang et al., 2025; Liu et al., 2025a; Xu et al., 2025; Liu et al., 2024a). Several benchmarks have been developed to measure hallucination, including Causal-HalBench (Xu et al., 2025) and BEAF (Ye-Bin et al., 2025). Prior work generally attributes hallucination to spurious correlations and language prior override (Ye et al., 2025; Liu et al., 2025a; Xu et al., 2025; Carragher et al., 2025).

Most existing work focuses on weak spurious relationships, such as a skiing scene without a human Ye-Bin et al. (2025), a bear on skis Xu et al. (2025), a tennis ball play without the ball He et al. (2025), or a juice stall with no cups Liu et al. (2025a). Less attention has been paid to objects with an essential part missing, such as an airplane without engines. Additionally, prior studies have primarily concentrated on smaller VLMs such as LLaVA Liu et al. (2024b; 2023), OpenVLThinker Deng et al. (2025), and similar models. It is not surprising to see these models make mistakes (that SOTA models won't), whereas our work targets SOTA large-scale VLMs.

The work that focused evaluation primarily on smaller models (e.g., 7B–13B scale) may not adequately represent the capabilities of production-grade VLMs. While such evaluations can reveal limitations in smaller models, they risk overstating the severity of issues that may be less pronounced or qualitatively different in frontier systems. Our focus on SOTA models ensures that observed failures reflect genuine limitations of current best practices rather than artifacts of model scale.

Since our goal is to characterize a particular failure mode in capable vision-language systems rather than to measure general model competence, we restrict our analysis to models where this distinction is meaningful. Sample results from smaller models are included in the appendix for reference.

In the image generation literature, VSF Guo & Du (2026) observed that generating such images is itself difficult for models. Their method improves the generation of objects with missing essential parts, but when evaluating the outputs, they found that VLMs sometimes hallucinate the missing component even when the generative model successfully omitted it. Nevertheless, the LLM still shows high agreement with humans, likely due to the missing objects are relatively obvious. Additionally, they finetuned their own model, NegAwareQwen, which shows much less hallucination.

Objects with missing parts present a unique challenge for VLMs, stemming from both real-world knowledge bias (e.g., the expectation that airplanes must have engines to fly) and the scarcity of such images in training data. Despite this, the problem has clear practical value, both for evaluating image generation models as in VSF and for real-world monitoring and inspection tasks. Our mini-benchmark (MissingBench-Verified) shows that current models consistently assert that a removed object is still present. Notably, even with external assistance such as image processing tools and a simulated perfect object detector confirming the object's absence, model performance remains below an acceptable level.

Figure 1: Vision language models could have severe hallucinations that a missing element of an object is still there, refuse to believe an external tool, and let internal prior knowledge overwrite visual context.

Across these hallucination studies (Xu et al., 2025; Ye et al., 2025; Ye-Bin et al., 2025; Liu et al., 2025a), a consistent finding is that current vision-language models often default to their learned priors or internal knowledge, even when it contradicts the prompt or visual input. In the MMKC-Bench's (Jia et al., 2025) evaluation, models were generally able to recognize that a conflict was present, but still "tend to favor internal parametric knowledge over external evidence" when answering. In other words, if the user's instruction or an external document stated something that clashed with the model's own training (or with the image), the model often trusted its internal belief. This might seem harmless or beneficial for trustworthy AI, as it prevents the model from being misled by misinformation, but it is actually the other way around, as AI's internal knowledge could be wrong, outdated, or biased (whether naturally from training data or intentionally from developers (Guo et al., 2025c)). Another work, Insight Over Sight (Liu et al., 2025b), found that when showing an image that conflicts with its internal assumptions, they observed an "overreliance on parametric knowledge" in about 20% of queries, where the model's answer ignored the visual scene in favor of what it expected or assumed instead. In the SegSub (Carragher et al., 2025) analysis, models showed uneven robustness: they were relatively resistant to simple parametric knowledge conflicts (only about a 20% chance of following a false prior fact inserted into an image), but struggled with counterfactual or complex conflicts – e.g., correctly identifying an unrealistic condition that happened only $< 30\%$ of the time, and resolving conflicts between multiple sources was almost never done correctly ($< 1\%$ accuracy). They found that the model will mostly rely on the image and not the internal truth.

Even when using tools, when the LLM receives conflicting information from tools and internal knowledge or observations, it might still prefer their own "beliefs" rather than those of the tools. This could explain some abnormalities in users' reporting, where the LLM refuses to believe the actual datetime, numerical calculations, or recent dramatic events in real life, even if the tools are providing accurate information.

Formally, this has been observed in ClashEval (Wu et al., 2025) where the model could use the external evidence if its initial confidence is weak. However, models could still stick with their internal beliefs if the context is too ridiculous. Similar to above, this is helpful if the information the model stored is correct, but it could lead to unwanted responses if the model is confidently wrong, which could happen if most of its training data is outdated. Tug-of-War (Jin et al., 2024) finds that " even when provided with correct external evidence, ChatGPT often persists in trusting its incorrect internal memory for more than half the time." A more recent study (Cheng et al., 2026) finds that when there is a tool-memory conflict, none of the existing approaches can effectively resolve these conflicts.

## 2 Methods

### 2.1 Dataset Construction

We construct the dataset using real-world captured photos and internet images, each collected alongside the name of the element to be removed. These images are not scraped using automated tools but are rather human-chosen images with high selectivity. Elements are removed using two approaches. The majority of images are processed with Nano Banana 2, a SOTA image-editing model, while the remaining images are edited manually using the image eraser tool `cleanup.pictures`. All images undergo human verification to filter out failed or unnatural edits. In total, we have 118 images in the dataset. The dataset is relatively small, but it is intended to show the existence of the problem rather than to qualitatively benchmark current models. That is why we named it MissingBench-Verified. A word cloud of the missing objects are shown in Figure 2.

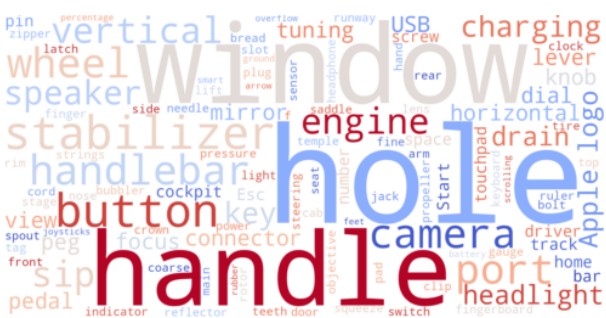

Figure 2: A word cloud of missing objects of our dataset.

### 2.2 Evaluation and Metrics

We evaluate 5 popular proprietary models and 5 SOTA open-source models, with names listed in Table 2. All models are queried via OpenRouter with default settings. Each model is prompted to produce a reasoning process before giving a final answer, even for non-reasoning models, as CoT is beneficial for non-reasoning models (Wei et al., 2023). The query is formulated as a multiple-choice question asking whether the target item is visible in the image, with options: "it is clearly visible," "it is visible but hard to see," and "it is not visible." An example query is shown in Figure 1. Structured output is used to enforce a JSON response format that contains both the reasoning process and the final answer.

We first evaluate all models on original, unedited images where the queried item is present, establishing a baseline for model response behavior. For edited images, we test four settings:

- **Direct query**: The model is directly queried about whether the target object is present. This tests whether the model can independently detect that the item has been removed.

- **Pre-answer tool injection**: The query includes a simulated object detection result injected into the model's context before it responds. The simulated result always returns undetected with low confidence, representing a perfect detector (i.e., successfully avoiding hallucinating the object; low confidence means it did not detect the object being present). This tests whether external tool output can correct model responses and serves as an upper bound for tool-augmented approaches.

- **Post-answer tool injection**: Similar to the previous setting, but the simulated detection result is injected after the model has already produced an initial response. This tests whether models exhibit confirmation bias and resist updating their answers.

| Model | Direct Query | Think with Images |
|-------|:------------:|:-----------------:|
| Claude Sonnet 4.6 | 98.3 | - |
| GLM 4.6V | 94.1 | - |
| GPT-4o | 93.2 | - |
| GPT-5.2 | 93.2 | - |
| Gemini 3 Flash Preview | 96.6 | 98.3 |
| Gemini 3.1 Pro Preview | 94.9 | - |
| Kimi K2.5 | 95.8 | - |
| Llama 4 Maverick | 94.9 | - |
| Qwen 3.5 Flash | 95.8 | 93.2 |
| Qwen 3.5 Plus | 94.1 | 96.6 |

Table 1: Results for each model on original images, showing the percentage of trials where the model answered 'Visible' or 'Hard to See' (which are correct answers). Models marked with - were not tested with image processing tools due to compatibility and cost constraints.

- **Thinking with images**: The model is given access to basic image processing tools, including cropping, brightness and contrast adjustment, and binarization, which it can apply before giving a final response. This tests whether processing the image, such as zooming in on a region of interest, can mitigate model hallucination. Since this requires the model to have think-with-images ability and is time-intensive, we only tested Gemini 3 Flash, Qwen 3.5 Plus, and Qwen 3.5 Flash.

We also tested how the reasoning effort setting could affect models' performance. We selected GPT-5.2 to test it under different reasoning efforts, including chat mode (no reasoning), minimal, low, medium, high, and xhigh. For comparison with the traditional object detection model, we also included Owlv2 (Minderer et al., 2024) for comparison, as Owlv2 has been shown to demonstrate high-performance in challenging situations (Guo et al., 2025b;a) and LLM-Det is a recent strong open-vocabulary detection model. For OWLv2, since the output is threshold-based, we run the model with a minimal threshold (0.05) and compute the average precision (AP) without the IoU constraint (using the highest score box), as testing localization is not our goal, and the AUROC. We also report the maximum achievable accuracy ($A_{max}$) by selecting the optimal threshold on the test set, following standard practice for threshold-based detectors, and serves as an upper bound, and the results under the default general purpose threshold. For the direct query setting, we also tested the NegAwareQwen-27B from the VSF paper.

## 3 Results

### 3.1 Results of VLMs

To verify that these models can successfully detect the objects in the original image, we first tested all models on the original image in a direct query setting and think with image processing tools setting. Results are shown in Table 1. We can observe that in original images, the models are able to correctly identify these objects being in the image.

The evaluation of ten leading VLMs on images where essential components were removed reveals a widespread susceptibility to visual hallucinations. As shown in Table 2, all tested models failed to achieve an accuracy (correctly identifying the object as "Not Visible," or NV) higher than 75%. Nearly half (4 out of 10) of the models identified the absence in less than 50% of the cases.

The lowest accuracy was recorded by Claude Sonnet 4.6 at 44.1%. Notably, models like Qwen 3.5 Flash and GLM 4.6V exhibited a strong "existence bias," frequently asserting that removed objects were "Clearly Visible" (V, i.e., answer 0). In the case of Qwen 3.5 Flash, the model chose the V option in 51.7% of trials, which actually exceeded its correct NV detections (47.5%). Surprisingly, NegAwareQwen performed very poorly, likely due to having been finetuned for the specific cases in VSF, where the missing object is easy to spot, and thus failed in different settings, or due to its size being too small.

| Model | V | HS | NV |
|-------|-----|------|------|
| Claude Sonnet 4.6 | 16.1 | 39.8 | 44.1 |
| Gemini 3 Flash Preview | 27.1 | 2.5 | 70.3 |
| Gemini 3.1 Pro Preview | 22.9 | 1.7 | 75.4 |
| GPT-4o | 24.6 | 16.9 | 58.5 |
| GPT-5.2 | 25.4 | 16.9 | 57.6 |
| Qwen 3.5 Flash Team (2026) | 51.7 | 0.8 | 47.5 |
| Qwen 3.5 Plus Team (2026) | 29.7 | 22.0 | 48.3 |
| Kimi K2.5 Kimi Team (2026) | 32.2 | 9.3 | 58.5 |
| GLM 4.6V GLM-V Team (2026) | 46.6 | 0.8 | 52.5 |
| Llama 4 Maverick noa | 26.3 | 24.6 | 49.2 |
| NegAwareQwen-27B Guo & Du (2026) | 57.6 | 0.8 | 42.5 |

Table 2: Results for each Model on edited images. V, HS, NV means Clearly Visible, Hard to See, and Not Visible. Since the image is edited, "Not Visible" should be the correct answer. All model has an accuracy less than 75%, with the lowest being only 44%.

Building on this baseline, we examine whether external tool evidence can correct these failures. We inject a simulated detection result either before (pre-answer) or after (post-answer) the model's first response. In pre-answer injection, while Kimi K2.5 and Qwen 3.5 Plus/Flash showed some gains with $\Delta$ values ranging from 14.4% to 16.9%, overall performance remained low. The post-answer setting highlights the "confirmation bias" inherent in many models. While Qwen 3.5 Plus showed the largest improvement ($\Delta = 24.5\%$), other models showed minimal gain or even a slight decrease; given the small sample size, these decreases should not be over-interpreted and likely reflect noise. Despite the presence of a simulated "perfect" detector, most NV rates remained below 75%, with Gemini 3.1 Pro Preview being the sole exception at 82.2% in the post-answer setting. This suggests that hallucinations in most models are deeply embedded in their perception, resisting external correction. In other words, these models do not fail because they made a simple mistake; they fail because they are deep-down convinced the missing object is actually there.

We further ask whether granting models access to image processing tools (e.g., cropping, contrast adjustment) enables autonomous inspection to resolve these failures. According to Table 4, these tools provided minimal benefit. While Gemini 3 Flash Preview and Qwen 3.5 Flash achieved minor NV increases of 4.0% and 4.2% respectively, Qwen 3.5 Plus experienced a slight degradation of $-0.9\%$, also likely due to noise. This is surprising as previous works have shown that such tools can enhance models' reasoning abilities (OpenAI; Yang et al., 2026)[1] and on theory could decrease hallucinations by examining the images more carefully and in detail. We discuss a detailed failure analysis of why image processing tools did not help in Section **??**.

Finally, we assess whether increasing test-time compute can overcome these perceptual failures using GPT-5.2 or if more thinking can lead to worse hallucination, like observed in Liu et al. (2025a). As shown in Table 7, scaling "Reasoning Effort" from "None" to "xHigh" resulted in a small decrease in performance. Although the decreasing trend is consistent with previous findings that more thinking can cause more hallucination, this decrease is not statistically or practically significant. We attribute this to the nature of our test: prior work suggests that extended reasoning degrades performance by reducing attention to the image in favor of language-based inference or prior knowledge. In our setting, however, hallucinations appear to be triggered regardless of reasoning effort, suggesting they are deeply embedded in the model's parametric knowledge or visual processing rather than induced by overthinking. Similarly, it can also explain why the model's performance is not improved by thinking longer, either.

## 3.2 Results of Object Detection Models

We evaluated the object detection models OWLv2 (Minderer et al., 2024) and YOLOE (Wang et al., 2025). Because these methods rely on a decision threshold, we report their AP (without IoU constraints), AUROC, the maximum accuracy achieved under an optimal threshold, the accuracy obtained with a general-purpose

---

[1]Thinking with images improves the results most with corresponding post training, which Qwen 3.5 likely had

|  | Pre-Answer | | Post-Answer | |
| Model | NV | Δ | NV | Δ |
|---|---|---|---|---|
| Claude Sonnet 4.6 | 57.6 | 13.5 | 48.3 | 4.2 |
| Gemini 3 Flash Preview | 72.9 | 1.7 | 66.9 | -4.3 |
| Gemini 3.1 Pro Preview | 72.0 | -4.3 | 82.2 | 5.9 |
| Llama 4 Maverick | 61.9 | 12.7 | 52.5 | 3.3 |
| Kimi K2.5 | 72.9 | 14.4 | 76.3 | 17.8 |
| GPT-4o | 65.3 | 6.8 | 66.9 | 8.4 |
| GPT-5.2 | 64.4 | 5.9 | 64.4 | 5.9 |
| Qwen 3.5 Flash | 64.4 | 16.9 | 56.8 | 9.3 |
| Qwen 3.5 Plus | 66.1 | 16.9 | 73.7 | 24.5 |
| GLM 4.6V | 61.0 | 8.5 | 56.8 | 4.3 |

Table 3: Results with pre-answer injected and post-answer simulated detection results. Pre-answer injection means the tool call is injected before the model's first answer, and post-answer injection means injecting the tool result after the model's first response and asking it to respond again. We give a simulated perfect object detection result to the model. We can see a slight improvement in the results, but they are still mostly below 75%, with the exception of Gemini 3.1 Pro with post-answer injection at 82%. Δ is measured as the difference between injection results and original results.

| Model | V | HS | NV | ΔNV |
|---|---|---|---|---|
| Gemini 3 Flash Preview | 20.5 | 4.3 | 75.2 | 4.0 |
| Qwen 3.5 Flash | 40.7 | 7.6 | 51.7 | 4.2 |
| Qwen 3.5 Plus | 33.1 | 18.6 | 48.3 | -0.9 |

Table 4: Results with image processing tools. Δ is measured between the direct query and the image processing tools. We can observe that there are minimal improvements.

threshold (0.3), and the accuracy on edited images using a threshold that yields accuracy $> 80\%$ on the original images to align with the VLM performance setup. For OWLv2, we used `owlv2-large-patch14-ensemble` and for YOLOE we used `yoloe-26l-seg.pt` from Ultralytics. Results are shown in Table 6.

We observe that object detection models also cannot correctly determine if the targeted object is missing from the image better than VLMs do. However, by examining the images and the accuracy at original threshold $t_0$, we find that instead of hallucinating the objects as VLMs did, they exhibit different failure modes. The object detection models simply cannot reliably detect the target or lack understanding of it. For OWLv2, at the default threshold, it does a reasonable job avoiding false positives on missing objects, but also sometimes fails to detect the objects in original images. For YOLOE, it almost completely avoided the missing objects in edited images but also failed to detect them in original images.

For OWLv2, when pushing the original accuracy higher, we have to lower the threshold to a point where the model becomes confused with basic object semantics. For example, as shown in Figure 3 bottom, the object detection model successfully avoided the removed drain hole; however, it misidentified the bubbler hole as a drain hole with a confidence score of 0.2. Note that 0.2 is not considered a low confidence score, as the official recommended thresholds include 0.1, under which the model would also yield false positives. This confusion arises because at lowered thresholds, semantically similar objects become harder to distinguish.

For YOLOE, on the other hand, it completely failed to detect some objects even at extremely low thresholds (0.01). At this threshold, the original image detection success rate is only around 70%. This suggests a lack of understanding for some objects in the model's knowledge or detection capability.

Given that this appears to be a performance limitation rather than a deeply embedded hallucination, it is likely more amenable to improvement than the VLM case. However, this also poses a question on how to balance a model's knowledge without making it stubborn to internal bias.

| Gemini-3-Flash | Qwen-3.5-Flash | Gemini-3.1-Pro | Best in Liu et al. (2024b) |
|---|---|---|---|
| 86.0% | 75.6% | 89.8% | 63.3% |

Table 5: Scores of RH-Bench on SOTA models

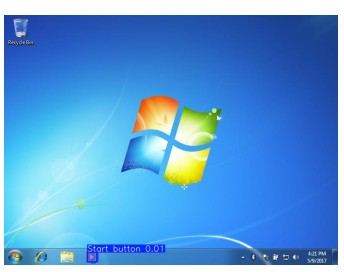 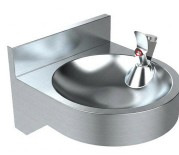

Figure 3: (Top) A false negative example from YOLOE, where it failed to detect the windows start button even at extremely low threshold (0.01) (Bottom) A false positive example from OWLv2. The target object is "drain holes" and OWLv2 detected the bubbler hole with confidence 0.2.

Critically, our work demonstrates that commonly proposed mitigation strategies—including external tool evidence, image processing capabilities, and extended reasoning—all fail to address this class of hallucination.

### 3.3 Comparison With Previous Work

As discussed in the introduction, several recent works have evaluated object hallucination in VLMs (Ye-Bin et al., 2025; He et al., 2025; Liu et al., 2025a). Our work differs in that we focus on missing *essential* parts, which poses a fundamentally more challenging test for VLMs. When an object with a weak association is absent (e.g., a bathroom without a toothbrush), the model can more easily acknowledge its absence, as such scenarios are plausible and likely present in training data. In contrast, an airplane without engines represents a highly implausible configuration that strongly violates the model's learned priors, creating a stronger conflict between visual evidence and parametric knowledge that triggers higher hallucination rates.

We view our benchmark as a form of adversarial stress testing. To validate that it poses a harder challenge, we evaluated Gemini-3-Flash and Qwen-3.5-Flash the RH-Bench from Liu et al. (2025a). Results in Table 5 show that modern VLMs achieve substantially higher accuracy on their benchmark compared to ours. This gap demonstrates that our dataset successfully identifies a more severe failure mode resistant to current model capabilities.

We focus our evaluation on frontier proprietary and large open-source models. We did not include smaller open-source models (3B–14B parameters) in our main evaluation. Initial experiments with smaller models (e.g., LLaVA 7B, OpenVLThinker 7B) revealed that their failures stem primarily from basic instruction following and language understanding rather than visual hallucination. For instance, LLaVA 7B frequently produced malformed outputs such as strings of random numbers instead of coherent reasoning, indicating fundamental competence issues. For cases where SOTA models fail to detect missing parts, smaller models similarly fail; for cases where SOTA models succeed, smaller models often still fail due to comprehension issues unrelated to the hallucination phenomenon we investigate.

## 4 Failure Case Studies

In most scenarios, the model simply hallucinated the presence of the object, and in the tool-injection evaluations, its most frequent error was to disregard the tools' outputs. Figure 1 illustrates an example that combines both of these failure modes.We observed fewer hallucinations of the explicit language-overwrite type (e.g., the object is not present, but the context strongly suggests it is), as reported in Liu et al. (2025a), in SOTA models, even though they are still common in smaller models. Nonetheless, we identified several interesting categories of failures.

| Metric | Owlv2 | YOLOE |
|---|---|---|
| AP | 0.745 | 0.680 |
| AUROC | 0.710 | 0.670 |
| Max Acc | 0.657 | 0.610 |
| Acc | 0.657 | 0.521 |
| Original Acc $t_0$ | 0.474 | 0.08 |
| Edited Acc $t_0$ | 0.838 | 0.974 |
| Original Acc $t_1$ | 0.805 | 0.695 |
| Edited Acc $t_1$ | 0.415 | 0.542 |

Table 6: Object detection results. Original accuracy and edited accuracy $t_0$ refers to the accuracy at the default threshold. Original accuracy $t_1$ refers to the accuracy at the specified threshold, where it exceeds 80%, and edited accuracy $t_1$ refers to the accuracy on edited images with this threshold. The YOLOE model has a very poor detection success rate using default thershold, likely due to the thershold is default for all YOLO model (0.25) and not tuned for this specific version.

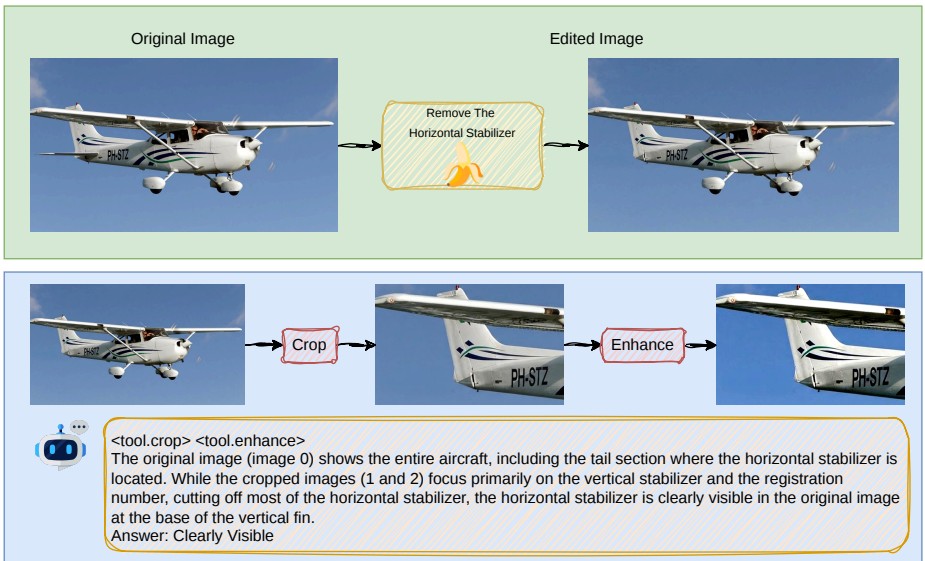

Figure 4: An interesting failure case arises in the "with image processing tools" setting: the LLM correctly uses the cropping tool, and the resulting cropped image clearly shows that the horizontal stabilizer is missing. However, instead of acknowledging this absence, the model attributes it to a cropping error ("cropped out"), rather than concluding that the horizontal stabilizer is not present.

**Cropped out or missing?** Figure 4 shows an interesting failure case in which the LLM distrusts the results of its own tool calls. The horizontal stabilizer (the horizontal "tail") has been removed using Nano Banana 2. The model is instructed to use tools to analyze the image, and it correctly applies a cropping tool followed by a contrast-enhancement tool. The crop properly isolates the tail section where the horizontal stabilizer should appear, and the full relevant region is contained within the frame. The cropped image clearly indicates that the horizontal stabilizer is absent. However, instead of inferring that the component is missing, the model blames its absence on the cropping process (i.e., it assumes the part was cropped out) and then refers back to the original image to claim that the horizontal stabilizer is present.

This pattern is consistent with other cases in which the model correctly notes that an object does not appear in the cropped region but attributes this to the object being outside the frame rather than absent, even when the primary object is fully visible and the expected location of the missing component falls clearly within

| Reasoning Effort | V | HS | NV |
|---|---|---|---|
| None | 26.3 | 8.5 | 65.3 |
| Low | 25.4 | 8.5 | 66.1 |
| Medium | 27.1 | 12.7 | 60.2 |
| High | 28.0 | 14.4 | 57.6 |
| xHigh | 20.3 | 22.0 | 57.6 |

Table 7: Results for GPT-5.2 at different reasoning effort. Although there is a difference, the proportions z-test p-value is 0.174 between the largest difference group (Low vs. xHigh) for "not visible", which is not significant. Note that this being not significant strength our point: the halluciniation is embedded into the system and does not change much with reasoning efforts.

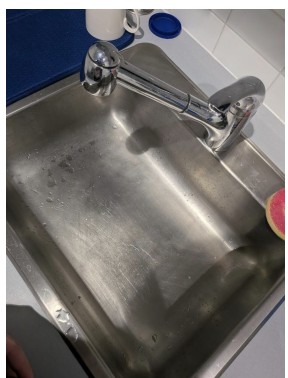 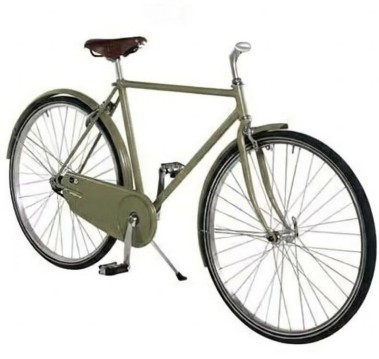 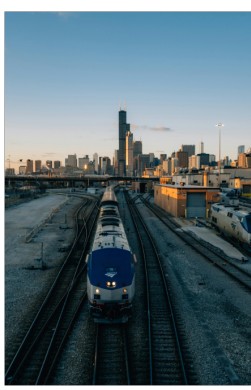

Figure 5: **(Left)** A "successful failure" case where the model correctly said the drain hole is missing but attribte it to out of frame or obstruction. The model's response is "The photo shows the sink basin and faucet, but the typical circular drain opening is not seen anywhere on the basin floor or near the bottom edge; it appears to be out of frame or obscured." **(Middle)** A case of pedantic defination, the bike handle is clearly removed with only the stem still attached on the body, the AI strength the deination of handle and argued that the stem part is the handle. **(Right)** "It should have it" thinking failure case. The model responsed with "The image shows a train on ⋯ The image does not provide a close-up view of the driver's area, but based on typical train designs, the driver window should be visible as part of the front view. Answer: Clearly Visible"

the image. One of these examples is shown in Figure 5, where a human can clearly see that the drain hole is missing and where it should appear is within the frame. The AI asserts it is not visible due to being out of frame or obscured. One possible explanation is that the model processes images as sequences of tokens and lacks robust spatial reasoning to distinguish between an object lying outside the frame and an object that is simply not there.

**Pedantic Redefinition.** A few edited images could technically cause ambiguity if one stretches the definition, though a reasonable person would not interpret them this way. This is similar to arguing that humans can survive without drinking water because they drink apple juice—technically correct but absurd, a pattern known as "definitional retreat." In our cases, models exploit this ambiguity to rationalize the object's presence.

An example is shown in the middle of Figure 5. The handlebar grips on both sides have clearly disappeared, leaving only the stem attached to the bicycle body. For a reasonable person, this bike does not have a handlebar. Yet the model argues that this stem itself constitutes the handle: "The bicycle's handlebar area is visible on the right side near the front wheel: a silver handlebar/stem with a grip is clearly shown above the front fork." This behavior could stem from either genuine visual misunderstanding or the model's attempt to justify its prior belief that handles should be present.

**"It should have it" thinking** Even though this is rare, another type of error, similar to the ones observed in Liu et al. (2025a), exists in our testing. It happens when the LLM asserts that an object should be there based on common facts or explicit language prior. An example is shown in Figure 5 as the leftmost sample. The AI is a bit confused about whether the window is visible, but then it defaults to what a typical train would look like. This type of error is mainly observed in older models like LLaMA 4 and GPT-4o.

## 5 Limitations and Furture Work

The main limitation of our study is the relatively small size of the dataset. However, our objective is to illustrate that the problem exists, rather than to establish an exact leaderboard-style ranking of models. Future work using an automated data pipeline could generate a much larger dataset. Another limitation is that we did not explore hallucination-specific training-based mitigation strategies. These approaches require model training, which is impractical for two reasons: we lack a separate training set, and training state-of-the-art models is nearly impossible. Furthermore, we aim to evaluate models in their default configurations, with minimal intervention or modification to the models themselves.

## 6 Conclusion

We present MissingBench-Verified, a benchmark designed to evaluate a specific and practically relevant hallucination scenario: when vision–language models (VLMs) fail to recognize that an essential component of an object has been removed. Across ten leading models, we observe consistent and significant failure rates that persist even when external tool evidence explicitly contradicts the model's visual perception. Notably, our benchmark elicits substantially higher error rates compared to prior hallucination benchmarks, demonstrating its effectiveness as a stress test for VLM robustness. Error analysis reveals that models frequently dismiss tool outputs, attribute missing regions to occlusion or framing artifacts, or confabulate object attributes to reconcile the contradiction. These behaviors suggest that the hallucinations are not merely superficial reasoning errors, but instead reflect deeply rooted priors embedded in the models' parametric knowledge. Critically, we find that existing mitigation strategies—including tool-assisted verification, autonomous visual reasoning, longer reasoning durations, and fine-tuning on an easier dataset—provide negligible improvement, indicating that this failure mode cannot be addressed through current prompting or post-hoc correction techniques. Our findings highlight a fundamental limitation of current VLMs for inspection and monitoring tasks and underscore the need for architectural or training-level interventions that enable models to override internal expectations when confronted with contradictory evidence.

## 7 An Interesting Note

While probing this idea with an LLM, we noticed an interesting side effect: the model criticized our dataset for supposedly containing imperfect removals, insisting that the target objects were still partially visible even when they had been completely erased. In other words, the model hallucinated their presence. This suggests that hallucinated objects might function as implicit signals for detecting LLM-written reviews, analogous to how models sometimes treat names like "GPT-5.2" or "Gemini-3" as fake version numbers and accuse the paper of being fabricated. Any reasonable human with real-world experience would know that these models actually exist, whereas an AI model, fixed at a particular training cutoff, cannot access current reality.This is similar to the way one top 2026 conference's organizers were rumored to have embedded special markers into submitted PDFs in order to detect reviews written by LLMs..

Taking this further, we speculate that this behavior could be exploited as a novel attack vector. Since an LLM may consistently perceive an object that does not exist in an image, such images could serve as a form of visual prompt injection or adversarial attack; combined with a pre-set textual prompt condition, they could trigger harmful behavior in a VLLM-based agent that a human operator would not anticipate, as the triggering object is invisible to humans. Even though this might be hard to exploit in real-world settings, it is an interesting direction or basis for more advanced attacks.

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
