# OpenReview forum: "MissingBench-Verified: Probing Vision-Language Models’ Inability to Detect Missing Object Parts"
_TMLR — Withdrawn by Authors_

### Review · Reviewer_ppH8 · 2026-04-07

**Summary Of Contributions:**

Note that this paper doesn't clearly state their contribution and the following is summarized by the reviewer.
1. The paper constructs a small but carefully verified benchmark targeting a specific VLM failure mode: hallucinating essential object parts that have been physically removed from images.
2. They test SOTA models and find that all models fail substantially, with no tested mitigation strategy providing meaningful improvement.

**Audience:**

Yes

**Audience Explanation:**

1. The failure mode is real and underexplored relative to standard hallucination benchmarks.
2. The multi-condition design (tool injection, post-answer tool injection, think with images) provide diverse experiment results.

**Claims And Evidence:**

No

**Claims Explanation:**

1. The benchmark comprises only 118 images, which is insufficient to establish that the identified failure mode is both widespread and critical. With such a small test set, the coverage of object categories, viewpoints, and editing conditions is inevitably limited. It is unclear whether the observed failure rates generalize beyond the specific scenarios curated here, or whether they reflect only this particular sample. The authors frame this as a fundamental and practically significant limitation of frontier VLMs, but a dataset of this scale can at best demonstrate that the problem exists in some cases.
2. Several of the highlighted failure cases in Section 4 are arguably not failures, or at least controversial. In Figure 5 (left), the drain hole may genuinely be at the edge of the frame or obscured given the shooting angle, thus the model's response that it is "out of frame or obscured" is a reasonable visual interpretation, not necessarily a hallucination. In Figure 5 (middle), the remaining stem is functionally graspable and could reasonably be called a handle under a broader but defensible semantic definition. Given the already small dataset, even a handful of mislabeled or ambiguous cases could meaningfully distort the reported failure rates and the qualitative conclusions drawn from them.
3. All images in the benchmark are processed by Nano-Banana, and while the authors apply human verification, it is unclear whether the edited images are truly indistinguishable from real photographs. Since the editing process can easily introduce domain gap, the observed hallucinations may partly due to the domain gap.

**Requested Changes:**

1. Please clearly write the summary of contributions in this paper
2. Provide inter-rater agreement statistics for the human verification step, and explicitly validate that each image's missing part is unambiguously within frame and clearly defined.
3. The prevalence of each failure mode should be reported, not just illustrated with examples.
4. Expand the dataset before claiming the failure mode is critical and widespread.

---

### Review · Reviewer_37GE · 2026-04-07

**Summary Of Contributions:**

The paper proposes a benchmark dataset comprising of images with missing parts, as a way to evaluate the performance of the state of the art VLMs on detecting whether the parts are missing or not. Overall, the paper looks promising- discovering some interesting observations such as the rigidity of the model in sticking to its priors even when opposing facts are presented as evidence. However, the lack of extensive experiments, loose problem definition and small benchmarks limit the significance of the paper. In its current state, the paper is best positioned as a workshop paper than a journal article.

### Strengths:
- The idea of creating a benchmark data that highlights fundamental issues with VLMs is important for the community, and if built the right way can help track progress.
- Clear and thorough discussion of related methods

### Weaknesses:
- Weak problem definition: The paper needs to clearly define the objects and the parts that they are operating with, and characterize them with difficulty levels. The current working definition is "essential component" but that is very vague and subjective.
- Very limited benchmark: Since the core contribution is a benchmark dataset, it needs to be much more extensive than what it is currently. With just ~100 data points, it is difficult to make any meaningful comparisons and conduct ablations. The difference in the results would more likely be down to noise than features.
- Dataset characterization: As a benchmark paper, the paper needs to have more discussion of the characteristics of the dataset. The only place where this is discussed currently is Figure 2 word cloud, but a word cloud doesnt highlight any quantitative (or even qualitiative) characteristics of the data. I advise the authors to study other leading benchmark papers and adopt their methods.
- Weak experimentation: There is absolutely no discussion of what were the prompts that were provided to the VLMs, how these could be engineered to invoke different behaviors, and how context engineering such as providing contrastive images can help the VLMs in mitigating hallucinations.

Additional comments:
- What was the basis of choosing the models that the paper is benchmarking? These should ideally be chosen systematically and cover the landscape.
- Minor typos (like missing references as ??, punctuation errors) in the paper that need to be fixed.

**Audience:**

Yes

**Audience Explanation:**

With how VLMs are being widely adopted, there is a need for the community to know their weaknesses, and so the paper definitely is of wide interest.

**Claims And Evidence:**

No

**Claims Explanation:**

See weaknesses

**Requested Changes:**

- More concrete problem definition
- A much more extensive benchmark dataset with proper characterization
- Study of how prompt and context engineering affects the hallucination issue

---

### Review · Reviewer_kEub · 2026-04-13

**Summary Of Contributions:**

This paper introduces MissingBench-Verified, a small benchmark (118 images) designed to test whether vision-language models (VLMs) can detect that an essential part of an object has been digitally removed (e.g., an airplane without engines, a phone without a home button). The authors evaluate 10 SOTA VLMs and 2 object detection models across four settings: direct query, pre-answer tool injection, post-answer tool injection, and thinking-with-image-processing-tools. They also test varying reasoning effort levels on GPT-5.2. The main finding is that all models fail frequently (best accuracy ~75% on direct query, with most under 60%), and none of the proposed mitigations (tool injection, image processing tools, extended reasoning, fine-tuning on an easier dataset) substantially resolve this. The paper includes a qualitative failure analysis categorizing errors into "cropped out vs. missing," "pedantic redefinition," and "'it should have it' thinking."

Key strengths:
(1) The problem is practically relevant for inspection or monitoring applications.
(2) The experimental settings are thoughtfully designed, particularly the pre/post tool injection paradigm that tests confirmation bias.
(3) The qualitative failure case studies (Section 4) are interesting and well-illustrated.

Key weaknesses:
(1) Limited novelty — detecting missing parts is closely related to well-studied anomaly detection and VLM hallucination benchmarks.
(2) The dataset is very small (118 images) with no statistical rigor (confidence intervals, significance tests across models).
(3) The paper lacks essential quantitative results, visualizations, and ablations to fully support its claims.
(4) The writing quality and organization need improvement.

**Additional Comments:**

The core observation that VLMs hallucinate the presence of missing essential parts and resist correction is
  interesting and practically relevant. However, in its current form, the paper reads more like an extended technical
  report or blog post than a rigorous research contribution. The dataset is too small, the analysis too shallow, the
  related work too narrow (missing the entire anomaly detection field), and the claims too strong for the evidence
  provided. The qualitative failure analysis (Section 4) is the strongest part of the paper and could be expanded. I
  encourage the authors to address the critical issues above and resubmit.

**Audience:**

Yes

**Audience Explanation:**

The topic of VLM hallucination is of broad interest to the TMLR community, and the specific angle of missing-part detection has practical relevance for inspection, quality control, and/or monitoring applications. The observation that models resist correcting their beliefs even when presented with contradictory tool evidence (the tool injection experiments) is a timely finding given the increasing deployment of tool-augmented VLM agents. However, the interest is contingent on the findings being supported by sufficiently rigorous methodology, which is currently lacking.

**Broader Impact Concerns:**

N.A.

**Claims And Evidence:**

No

**Claims Explanation:**

The paper's central claim that missing-part detection represents a "fundamental limitation" requiring architectural interventions, is not adequately supported by the evidence presented. Several specific issues below:

  1. Insufficient dataset size and no statistical analysis. With only 118 images, the reported accuracy differences
  between models (e.g., 44.1% vs. 75.4%) and between conditions (e.g., delta values in Table 3 ranging from -4.3% to
  +24.5%) lack confidence intervals or proper significance testing. The authors themselves acknowledge that small
  decreases "should not be over-interpreted and likely reflect noise" (Section 3.1), yet they draw strong conclusions
  from similarly small improvements. The one significance test reported (Table 7, p=0.174 for reasoning effort) actually
   undermines their argument, as it shows no significant effect, which the authors spin as supporting their point, but it
  could equally reflect insufficient statistical power from the small sample.

  2. Missing critical results and analyses. The paper lacks several essential components:
a. No per-category or per-object-type breakdown. Are failures uniform across object types (vehicles, electronics, plumbing fixtures) or
  concentrated in specific domains? The word cloud (Figure 2) hints at diversity but no stratified analysis is provided.
b. No confusion matrix or detailed error distribution beyond the V/HS/NV trichotomy.
c. No analysis of image edit quality. We don't know if models are detecting editing artifacts or hallucinating.
d. The "thinking with images" setting (Table 4) is tested on only 3 of 10 models, making it hard to draw general
  conclusions.
e. No human baseline. How well do humans detect these edits?

  3. The connection to anomaly detection is entirely ignored. Detecting that an expected component is missing from an
  object is essentially a visual anomaly detection task. The industrial anomaly detection literature (e.g., MVTec AD [1],
  PatchCore [2], WinCLIP [3], and other zero-shot anomaly detection methods) has extensively studied detecting missing,
  misaligned, or defective parts. The paper does not cite, compare with, or even acknowledge this body of work. This is
  a significant omission that undermines both the novelty claim and the claim that this is an unsolvable problem.

  4. Many other approaches remain untested. To name a few, few-shot prompting with negative examples, visual grounding methods, reference-image comparison, anomaly-detection-specific fine-tuning, etc.

[1] P. Berg TheCVFmann, M. Fauser, D. Sattlegger, and C. Steger, "MVTec AD — A comprehensive real-world dataset for unsupervised anomaly detection," in Proc. IEEE/CVF Conf. Comput. Vis. Pattern Recognit. (CVPR), Jun. 2019, pp. 9592–9600.
[2] K. Roth, L. Pemula, J. Zepeda, B. Schölkopf, T. Brox, and P. Gehler, "Towards total recall in industrial anomaly detection," in Proc. IEEE/CVF Conf. Comput. Vis. Pattern Recognit. (CVPR), 2022, pp. 14298–14308.
[3] J. Jeong, Y. Zou, T. Kim, D. Zhang, A. Ravichandran, and O. Dabeer, "WinCLIP: Zero-/few-shot anomaly classification and segmentation," in Proc. IEEE/CVF Conf. Comput. Vis. Pattern Recognit. (CVPR), Jun. 2023, pp. 19606–19616.

**Requested Changes:**

1. Substantially expand the dataset or provide rigorous statistical analysis. 118 images is insufficient for a
  benchmark paper. Either scale the dataset to at least 500+ images using the described pipeline or provide bootstrap confidence intervals for all reported metrics and proper paired significance tests. The current presentation mixes noise with signal.
  2. Add a human baseline. E.g., recruit annotators to perform the same task on the edited images. This is essential for
  establishing that the edits are convincing enough that the ground truth is unambiguous, and that the human-VLM gap quantifies the severity of the problem.
  3. Discuss and compare with the anomaly detection literature. At minimum, cite relevant work (MVTec AD benchmark,
  zero-shot anomaly detection with CLIP-based methods like WinCLIP, April-GAN, etc.) and discuss how this task relates
  to anomaly detection.
  4. Provide per-category analysis. Break down results by object type, size of missing component relative to the full
  object and edit method (tool vs. manual). This would reveal whether failures are uniform or concentrated, which is critical for understanding the phenomenon.
  5. Address image edit quality control. Provide evidence that the edits are undetectable to humans (or quantify their
  quality). If a model's "hallucination" is actually detecting a subtle inpainting artifact and then confabulating a
  reason for the object's presence, that is a fundamentally different failure mode than what the paper claims.

---

### Note · Authors · 2026-04-13

I have read and agree with the venue's withdrawal policy on behalf of myself and my co-authors.